# Virtual Care in Patients with Cancer: A Systematic Review

**Simron Singh [1], Glenn G. Fletcher [2] , Xiaomei Yao [2,3] and Jonathan Sussman [2,\*]**

[1]   Person-Centred Care, Ontario Health (Cancer Care Ontario), Sunnybrook Health Sciences Centre, Toronto, ON M4N 3M5, Canada; Simron.Singh@sunnybrook.ca

[2]   Program in Evidence-Based Care, Ontario Health (Cancer Care Ontario), Department of Oncology, McMaster University, Hamilton, ON L8S 4L8, Canada; gfletche@mcmaster.ca (G.G.F.); yaoxia@mcmaster.ca (X.Y.)

[3]   Department of Health Research Methods, Evidence and Impact, McMaster University, Hamilton, ON L8S 4L8, Canada

\*   Correspondence: sussman@hhsc.ca or ccopgi@mcmaster.ca; Tel.: +1-905-387-9495

**Abstract:** Virtual care in cancer care existed in a limited fashion globally before the COVID-19 pandemic, mostly driven by geographic constraints. The pandemic has required dramatic shifts in health care delivery, including cancer care. We conducted a systematic review of comparative studies evaluating virtual versus in-person care in patients with cancer. Embase, APA PsycInfo, Ovid MEDLINE, and the Cochrane Library were searched for literature from January 2015 to 6 August 2020. We adhered to PRISMA guidelines and used the modified GRADE approach to evaluate the data. We included 34 full-text publications of 10 randomized controlled trials, 13 non-randomized comparative studies, and 5 ongoing randomized controlled trials. Evidence was divided into studies that provide psychosocial or genetic counselling and those that provide or assess medical and supportive care. The limited data in this review support that in the general field of psychological counselling, virtual or remote counselling can be equivalent to in-person counselling. In the area of genetic counselling, telephone counselling was more convenient and noninferior to usual care for all outcomes (knowledge, decision conflict, cancer distress, perceived stress, genetic counseling satisfaction). There are few data for clinical outcomes and supportive care. Future research should assess the role of virtual care in these areas. Protocol registration: PROSPERO CRD42020202871.

**Keywords:** cancer; phone; systematic review; telehealth; telemedicine; teleoncology; videoconferencing; virtual care

## 1. Introduction

Virtual care, defined as interaction between a patient and clinician(s) that is not in-person, is also commonly referred to as remote care, telemedicine, or teleoncology, and is a subset of eHealth; the primary modalities are telephone and videoconferencing. Virtual care in patients with cancer existed in a limited fashion globally before the COVID-19 pandemic, mostly driven by geography constraints. While in-person care has been considered the "gold standard" of interaction between patients and physicians, there are several components of in-person care that may be delivered with equivalent effectiveness using non-in-person or virtual platforms. Evidence on virtual care is emerging but there remain numerous unknowns that need to be addressed to guide health care systems and cancer clinicians, as well as patients and caregivers, in understanding the potential for virtual care to substitute for in-person care. There are questions regarding efficacy and quality, as well as the system- and patient-level resources required.

There has been rapid adoption of virtual care due to the COVID-19 pandemic. This transition to virtual care has occurred in the absence of evidence as to its equivalency to traditional care. The objective of this systematic review was to find and evaluate clinical studies of virtual versus in-person care. The full review, including results of reports and

publications addressing technical requirements, equity, inter-professional care, and health-care provider compensation for optimal delivery of virtual cancer, is available on the Ontario Health (Cancer Care Ontario) website (https://www.cancercareontario.ca/en/guidelines-advice/types-of-cancer/68836), accessed on 24 August 2021.

## 2. Materials and Methods

### 2.1. Systematic Review Planning and Registration

A search of systematic reviews in PROSPERO, evaluation of known reviews and guidelines, and subject area knowledge of the Working Group members suggested that a systematic review specifically on this topic had not been published. We therefore designed and conducted this review, and the review protocol was registered on the International Prospective Register of Systematic Reviews (PROSPERO), CRD42020202871.

### 2.2. Literature Search Method

Embase, APA PsycInfo, MEDLINE, the Cochrane Library, and CINAHL were searched from 2015 to 1 August 2020 (Supplementary Material, Table S1). Clinicaltrials.gov was searched on 5 November 2020, for ongoing trials (Table S2), as well as systematic reviews and guidelines. While some systematic reviews or guidelines addressed aspects of the topic, they covered narrower topics, did not focus on cancer, missed several of the studies that we found, or were based on non-comparative studies, and are not discussed in this publication.

### 2.3. Search Strategy and Study Selection

This review included patients diagnosed with cancer who were undergoing treatment or follow-up. The comparison was virtual care versus in-person care between the patient and the same clinician (or team of clinicians).

Studies in which a subset of in-person visits were replaced by virtual visits were included. Studies had to be full-text primary publications in English or French with at least 30 patients per group. Non-comparative studies with more than 100 patients receiving virtual care were also included if they met all other criteria. Those excluded were trials that studied other interventions such as reduction in frequency of appointments; phone or text reminders; use of mobile or online apps, educational materials, or lifestyle adaptation; or replacement of in-person care from one professional or treatment team/unit (e.g., oncologist plus nurse) with virtual or in-person care by a different profession/team (e.g., community nurse or general practitioner). Publications of conference abstracts or other non-full text reports, editorials, opinions, comments or commentaries, notes, or news articles were also excluded.

Title and abstract screening were performed by one reviewer. In cases of uncertainty, the full working group determined inclusion or exclusion. Preferred (critical) outcomes were recurrence, survival, or other long-term objective outcomes. Patient experience outcomes, including acceptance of virtual care, symptoms, and quality of life, were considered important outcomes.

### 2.4. Data Extraction and Assessment of Risk of Bias

All included primary studies underwent data extraction by one reviewer, with subsequent independent audit of all extracted data. The risk of bias for randomized studies was assessed per outcome and per study using methods outlined in the Cochrane Handbook for Systematic Reviews of Interventions and the RoB2 tool [1,2]. The ROBINS-I tool was used for non-RCTs [3]. The risk of bias was performed by M.F. and G.G.F. independently and discussed with X.Y. to get consensus. The current review considered risk of bias, inconsistency, indirectness, imprecision, and publication bias (based on GRADE approach) in evaluating the quality of evidence [4].

*2.5. Synthesizing the Evidence*

Due to the large number of different study designs, interventions, comparators, patient populations, follow-up periods, and the outcome reporting time and methods involved, a meta-analysis or network meta-analysis was inappropriate to perform. Instead, the results of each study were presented individually in a descriptive fashion. Ratios, including odds ratios for dichotomous outcomes, were expressed with a ratio of <1.0 indicating a benefit for virtual care compared with in-person care. For continuous outcomes, mean differences or standardized mean differences were used as effect measures, and a two-sided significance level of $\alpha = 0.05$ was assumed.

### 3. Results

The literature search resulted in 11,307 citations, of which 216 required full-text review. Thirty-nine publications representing 23 clinical studies and 5 ongoing trials met our pre-planned selection criteria (Figure 1). The trials found have been divided into those that provide psychosocial or genetic counselling, and those that provide or assess medical and supportive care.

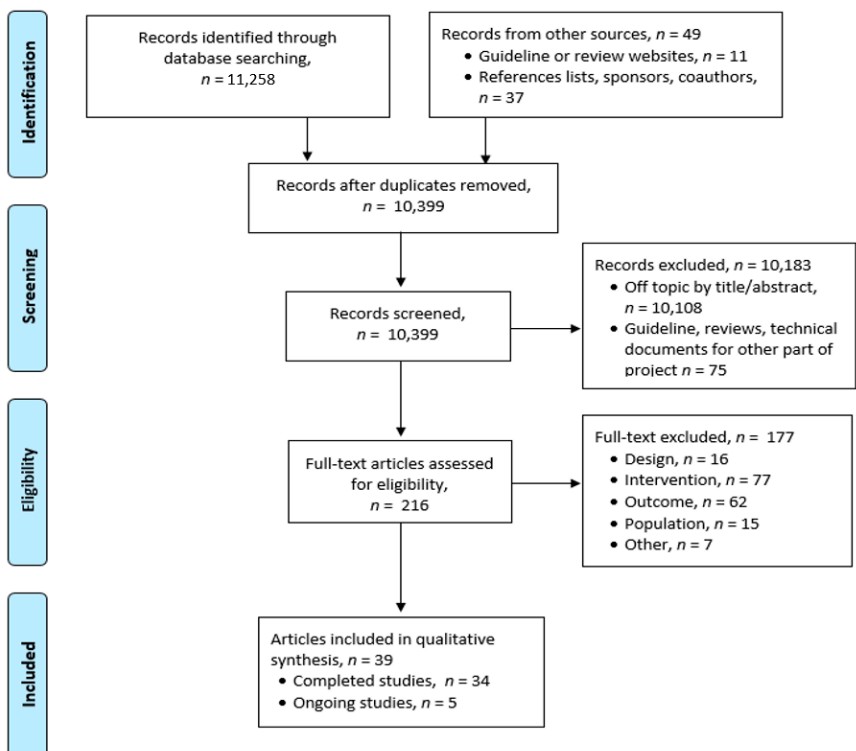

**Figure 1.** PRISMA Flow Diagram.

The overall risk of bias for randomized controlled trials (RCTs) was considered low for two studies and high for eight studies (Table S3). The risk of bias for non-RCTs ranged from moderate to critical risk using the ROBINS tool [3]. After also considering other domains (inconsistency, indirectness, imprecision, and publication bias), the overall certainty of evidence per outcome and per study was evaluated as moderate to very low for RCTs and very low for non-RCTs. Thus, we did not provide the risk of bias assessment table for non-RCTs.

*3.1. Counselling Studies*

Table 1 includes 16 publications of 10 studies, grouped as either general or genetic counselling [5–20]. The group psychotherapy RCT studied women with emotional distress after primary oncology treatment [5]. Emotional distress and post-traumatic stress symptoms improved in on-line counselling and in-person counselling groups; there were no

significant differences between groups. An RCT of cognitive behaviour in patients with cancer and high psychological needs for mental health compared telephone with face-to-face therapy [6]. Both arms had significant improvement in anxiety and depression, and equivalent improvement in stress and worry. They did not demonstrate full equivalence of the two arms, although both were effective and telephone care was non-inferior. The LEAN study compared telephone weight-loss counselling versus in-person counselling versus usual care in 100 breast cancer survivors [7]. Both groups improved, but there were no significant differences between study groups.

A survey of 209 cancer patients or their family members receiving psycho-oncology counselling using videoconferencing due to COVID-19 restrictions found that patients were grateful that care could be continued, but that it was more distant due to lack of non-verbal communication and non-recognition of signs of distress [8]. Therapists also missed non-verbal communication and informal physical contact and felt more exhausted; they were willing to continue videoconferencing if patients requested this, but preferred in-person sessions for more complex therapies. Approximately one-half of the patients indicated a preference to use video-consults a portion of the time when in-person care is again allowed.

The studies on genetic counselling included patients with higher risk for hereditary cancers and sometimes close family members. Counselling took place prior to genetic testing (and sometimes after), with the purpose of providing education and emotional support and helping patients to decide whether to undergo testing.

Two RCTs focused on rural communities and the third was conducted through cancer centres in four large cities [9–17]. They all found that virtual care via video or telephone was more cost-effective than in-person counselling. For the system, this was driven mainly by lower overhead (office space) and travel time for staff to attend remote clinics, while for the patient, it was due to less travel time and expense, and less time off work. The videoconference trial reported no difference in patient satisfaction [9]. The REACH trial reported that at one-year follow-up, telephone was not inferior to in-person counselling for all psychosocial and informed decision-making outcomes (anxiety, cancer-specific distress, perceived personal control, decisional conflict) [11–13]. The Jacobs et al. RCT reported that telephone was more convenient, resulted in no difference in knowledge or stress, but with less perceived support and emotional recognition [14–17]. There was no overall difference in patient satisfaction.

In the study by Mette et al., patients received in-person care if the genetic counselor or oncologist visited the regional extension centre on the day of their appointment; otherwise, they received counselling by video-teleconferencing at the regional centre [18]. There were no differences in satisfaction, feeling comfortable talking to the counsellor or listened to, having enough time, understanding information, finding the information to be valuable or that it helped to make health decisions, or whether they would recommend the program.

The non-randomized study by Solomons et al. compared videoconferencing for 106 patients in two remote sites versus counselling for 68 patients in an in-person clinic, with the grouping mainly determined by geographic proximity to the site [19]. Groups were unequal regarding age, personal history of cancer and type of cancer, and education. Knowledge of hereditary breast and ovarian cancer improved equally, and anxiety decreased in both groups. While videoconferencing reduced transportation needs and work absence, 32% of the patients indicated a preference for in-person care.

The study by Tutty et al. surveyed patients with high grade serous ovarian cancer who had received telephone genetic counselling [20]. Most patients were satisfied with the timing of the telephone call (97%) and the information provided (94%). Face-to-face counselling would have been preferred by 17% of patients, while 34% had no preference.

### 3.2. Medical or Supportive Care

As indicated in Table 2, the literature search found four RCTs, with one each for topics of pain management, solid tumour systemic therapy follow-up, endometrial cancer follow-up, and prostate cancer follow-up [21–38].

**Table 1.** General or genetic counselling.

| Author and Country | Patient Characteristics | Trial Type and Intervention | Sample Size (Patients per Group) | Outcome | Evidence Certainty |
|---|---|---|---|---|---|
| General counselling studies | | | | | |
| Lleras de Frutos et al., 2020 [5] | Adult women with cancer (81% breast cancer) and emotional distress after primary oncological treatment | Pragmatic RCT; Online videoconference vs. face-to-face group positive psychotherapy for cancer survivors; 12 weekly group sessions, online group had 11 sessions plus 1 in-person session | 269 of which 225 (108 vs. 117) randomized and 44 (16 vs. 28) selected their group | Measurement tools: HADS, PCL-C, PTGI; Emotional distress (anxiety and depression), post-traumatic stress symptoms improved in both groups, no significant difference between groups; Post-traumatic stress still above cut-off after treatment; No significant difference in attrition, integrity, or effectiveness after adjusting for baseline differences; Online counselling is not superior, both may be effective. | Very low |
| Watson et al., 2017 [6] UK | Cancer patients (except non-melanoma skin cancer) with high psychological needs for mental health and coping referred for psychological care | Equivalence RCT; Telephone-delivered cognitive behavioural therapy to face-to-face (treatment as usual) therapy; Median 4 sessions | 118 (60 vs. 58) randomized; 78 analyzed (43 vs. 35; including 5 vs. 6 who switched arms) | Measurement tools: HADS, MAC H/H, CLCC, CCQ, and additional post-therapy study-specific questionnaire; Both arms had significant improvement in anxiety, depression, HADS total score, cancer ($p < 0.01$); In-person but not telephone group had significant improvement in helpless/hopeless scale ($p = 0.13$ and $p = 0.015$); Stress and worry improved and were equivalent between groups. | Very low |
| Harrigan et al., 2016 [7] USA | Breast cancer survivors with BMI $\geq 25.0$ kg/m$^2$ diagnosed in the 5 years before enrollment with stage 0 to 3 breast cancer, who had completed chemotherapy and/or radiation therapy $\geq 3$ months before enrollment | 3 Arm RCT Telephone vs. in-person weight-loss counselling (11 sessions over 6 months) vs. usual care (brochures and referred to survivorship clinic, which offers a 2-session weight management program and an in-person counselling session) | 100 (34 vs. 33 vs. 33) 6-month data: 24 vs. 30 vs. 31 12-month data: 15 vs. 22 vs. 19 | Weight change at 6 months: 4.8 kg ($-5.4\%$), 5.6 kg ($-6.4\%$), 1.7 kg ($-2.0\%$); $p = 0.46$ telephone to in-person, $p = 0.009$ telephone to usual care; $p = 0.001$ in-person to usual care; Self-reported weight loss at 12 months was not significantly different between groups; Reduction in % body fat was significant for in-person ($p = 0.05$) but not telephone ($p = 0.37$) compared to usual care; Difference between telephone and in-person care was not significant ($p = 0.35$); Increased activity: $96 \pm 154$ min vs. $114 \pm 130$ min vs. $17 \pm 110$ min ($p < 0.05$); Change in number of steps per day: 948 vs. 1847 vs. $-330$ ($p < 0.05$). | Very low |

**Table 1.** *Cont.*

| Author and Country | Patient Characteristics | Trial Type and Intervention | Sample Size (Patients per Group) | Outcome | Evidence Certainty |
|---|---|---|---|---|---|
| General counselling studies | | | | | |
| Van der Lee et al., 2020 [8] Netherlands | Cancer out-patients or their family members | Single-arm study; Psycho-oncology counselling via video-consults instead of in-person; Survey of 34 psychologists and 2 psychiatrists giving video-consults | 239 surveys (209 patients, 30 therapists) | Patients reported being grateful for continued care, video-consults as more distant, harder for some to express feelings, missed travel time as time to prepare and process afterwards, missed having a place outside the home to leave their distress; Others found video from home as quieter and relaxed, with less stress due to travel and face-to-face contact; Therapists missed non-verbal communication and informal physical contact that normally helps release tension; Therapists had to work harder and felt more exhausted; All willing to continue video-consults if requested by patients; Preference for in-person sessions for more complex therapies. | Very low |
| Genetic counselling studies | | | | | |
| Buchanan et al., 2015 [9]; Datta et al., 2011 [10] USA | People referred to CGC and who preferred CGC locally instead of at the academic medical centre | RCT; Telegenetics vs. in-person; Same genetics counsellor for both groups; Not designed to test inferiority or equivalence of telegenetics vs. in-person counselling | 162 (81 vs. 81) randomized; 59 vs. 71 analyzed | Measurement tools: VSQ and GCSS 9GCSS0; Cost to health care system: $106 vs. $244/patient; Patient satisfaction was high and did not differ between groups; CGC attendance: 79% vs. 89%, $p = 0.03$; Lower computer comfort associated with lower attendance ($p = 0.02$); Of telegenetics group, 98% were comfortable with the system, but 32% would have preferred an in-person visit | Very low |
| Steffen et al., 2017 [11]; Kinney et al., 2016 [12]; Chang et al., 2016 [13] USA | Breast (91.5%) and ovarian cancer (8.1%) survivors at increased hereditary risk for BRCA1/2 mutations; history suggestive of HBOC meeting NCCN criteria for genetic counselling | RCT equivalency/ noninferiority trial; Telephone (TC) vs. in-person counselling (IPC) | 502 vs. 510 randomized; 464 vs. 437 eligible and received counselling; 402 vs. 379 analyzed [11]; 493 vs. 495 [12] | Groups did not differ at low levels of distress (27% vs. 30%) and risk (23.8% vs. 29.8%); At high distress uptake was 26.3% TC vs. 44.3% IPC (OR = 0.45, 95% CI = 0.27 to 0.76) and at high perceived risk uptake was 33.9% vs. 50.5% (OR = 0.50, 95% CI = 0.29 to 0.87); At 1-yr follow-up, TC was non-inferior to IPC for all psychosocial and informed decision-making outcomes (anxiety, cancer-specific distress, perceived personal control, decisional conflict); Telephone counselling cost: $120 (range $80−$200) vs. $270 (range $180−$400) per person. | Low |

**Table 1.** *Cont.*

| Author and Country | Patient Characteristics | Trial Type and Intervention | Sample Size (Patients per Group) | Outcome | Evidence Certainty |
|---|---|---|---|---|---|
| Genetic counselling studies | | | | | |
| Jacobs et al., 2016 [14]; Peshkin et al., 2016 [15]; Schwartz et al., 2014 [16]; Butrick et al., 2015 [17] USA | Women with BRCA 1/2-associated hereditary breast/ovarian cancer | Non-inferiority RCT; Telephone genetic counselling (TC) vs. usual care (UC; in-person) by trained genetic counselor; Follow-up telephone interview approximately 2 weeks later to assess perception and satisfaction with pre-test counselling | 669 (335 vs. 334) randomized; 554 (272 vs. 282) completed baseline and 2-week follow-up interview; 479 (236 vs. 243) analyzed | Measurement tools: BCGCKS, DCS, IES, MCS, PCS; TC noninferior to UC on all outcomes (knowledge, decision conflict, cancer distress, perceived stress, genetic counseling satisfaction); No difference and TC non-inferior in pre-test and post-test survey for satisfaction (83.1% vs. 86.8% very satisfied, $p = 0.22$), knowledge, and perceived stress; UC group had more decisional conflict but within non-inferiority bounds; TC was more convenient (OR = 4.78, 95% CI = 3.32 to 6.89) but with lower perceived support (52.9% vs. 66%, $p = 0.002$; OR = 0.56, 95% CI = 0.40 to 0.80) and emotional recognition (55.5% vs. 68.8%, $p = 0.001$; OR = 0.53, 95% CI = 0.37 to 0.76); 80.9% of TC preferred TC or had no preference; 84.2% of UC preferred UC or had no preference ($p = 0.3$); TC group had less uptake of subsequent BRCA 1/2 testing (84.2% vs. 90.1%; logistic regression model OR = 1.65, 95% CI 1.00 to 2.72); Genetic counsellor scores did not differ overall ($p = 0.910$); Scores did not differ by group, but were lower for minorities. | Moderate |
| Mette et al., 2016 [18] USA | Underserved primarily Hispanic population (95%); high risk based on their personal and/or family medical histories and meeting NCCN guidelines for genetic counselling | Non-randomized study; Telemedicine or video-teleconferencing vs. in-person; In-person appointments about once a month at each center (otherwise it was by video) and patients had no input on type of visit | 353 surveys, 119 responses (56 vs. 63) | There were no differences between the two groups for satisfaction, comfort talking, feeling listened to, enough time, understanding information, finding information valuable, information helpful to make health decisions, or likelihood of recommending the program. | Very low |

**Table 1.** *Cont.*

| Author and Country | Patient Characteristics | Trial Type and Intervention | Sample Size (Patients per Group) | Outcome | Evidence Certainty |
|---|---|---|---|---|---|
| Genetic counselling studies | | | | | |
| Solomons et al., 2018 [19] USA | New rural patients with personal or family history suggestive of HBOC susceptibility | Non-randomized study; Live-interactive videoconferencing from remote clinic vs. in-person; Groups matched by gender, race, health insurance status; Same counselor or oncologist for remote or in-person counselling; Questionnaire at pre-counselling, immediately after, 1 month after by mail, plus 4 weeks after test results for those undergoing genetic testing | 174 (106 vs. 68) 158 (90 vs. 68; 85% vs. 100%) returned pre- and post-counselling surveys; 65 (41 vs. 24; 46% vs. 35%) returned 1-month surveys | Measurement tools: 9 HBOC-related knowledge questions, PHQ-4, pre-validated survey; HBOC knowledge improved equally (evaluated only in patients with personal or family history of breast/ovarian cancer); Remote group had higher anxiety and depression pre-counselling; Decreased anxiety in both groups; Depression improved more in telegenetics group initially but was lower at 1 month in both groups; Telegenetics (remote) reduced transportation need and work absence; All patients satisfied with quality of care, 32% of remote patients noted preference for in-person care. | Very low |
| Tutty et al., 2019 [20] Australia | Women with high-grade serous ovarian cancer | Single-arm study; Telephone genetic counselling prior to testing and after testing if patient decides to be tested; Those with BRCA1/2 variant affecting function ($n = 26$, 9.2%) or significant family history requiring evaluation (e.g., Lynch syndrome) offered further in-person counselling | 284 counselled; 277 surveys; 107 responses (39%) | 40% had poor knowledge (<5/7 correct answers) for knowledge of hereditary breast and ovarian cancer syndromes; 97% satisfied with timing of telephone call and 94% satisfied with information provided; 17% would have preferred face-to-face counselling, 34% had no preference; Median per patient cost was AUD $91.52 telephone vs. $107.37 in-person. | Very low |

Abbreviations: BCGCKS = Breast Cancer Genetic Counseling Knowledge Scale, BMI = body mass index, BRCA1/2 = BReast CAncer gene 1 or 2, CCQ = Cancer Coping Questionnaire, CGC = cancer genetics counselling, CI = confidence interval, CLCC = Checklist of Cancer Concerns, DCS = Decisional-Conflict Scale, GCSS = Genetic Counselor Satisfaction Survey, HBOC = hereditary breast and ovarian cancer, HADS = Hospital Anxiety and Depression Scales, IES = Impact of Event Scale, NCCN = National Comprehensive Cancer Network, OR = odds ratio, MAC H/H = Mental Adjustment to Cancer Scale: Helpless/Hopeless subscale, MCS = Mental Component Summary, PCL-C = Posttraumatic Stress Disorder Checklist-Civilian Version, PCS = Physical Component Summary, PHQ-4 = Patient Health Questionnaire for Depression and Anxiety, PTGI = Posttraumatic Growth Inventory, RCT = randomized controlled trial, vs = versus, VSQ = Visit-Specific Satisfaction Questionnaire, yr = year.

**Table 2.** Studies except counselling.

| Author and Country | Patient Characteristics | Trial Type and Intervention | Sample Size (Patients per Group) | Outcome | Evidence Certainty |
|---|---|---|---|---|---|
| Kelleher et al., 2019 [21]; Winger et al., 2020 [22] USA | Patients with breast, lung, prostate, or colorectal cancer | Non-inferiority RCT; Videoconference vs. in-person psychosocial pain management; 4 sessions; Videoconference group given tablet (iPad) with data plan (also given to patients in the in-person group if needed to access website); Both groups had access to study website for self-assessment and to indicate preferences for content of next session | 178 (89 vs. 89) randomized; 137 (75 vs. 62) post-treatment; 128 (70 vs. 58) at 3-month follow-up | Measurement tools: CSQ and individual questions items; Similar patient burden and acceptability in both groups, but better feasibility in videoconference group; Pain severity, pain interference, physical well-being, physical symptoms, psychological distress, and self-efficacy for pain management all improved post-treatment compared to baseline in both groups, with continued improvement at 3-months only in the videoconference group; Videoconferencing was non-inferior at post-treatment, and at 3 months post-treatment; Completion of all sessions predicted improvement in pain severity, pain interference, and pain self-efficacy; Videoconference group was more likely to complete all sessions (83% vs. 65%, $p = 0.006$); Patients near medical center, with early cancer or less comorbidity, were more likely to complete in-person sessions. | Low |
| Walle et al., 2018, 2020 [23,24] Germany | Solid tumours and systemic therapy needing follow-up visit at outpatient clinic in 2–14 days, follow-up for 6 months | RCT; Mobile telephone vs. in-person visit | 66 (33 vs. 33) randomized; 48 (22 vs. 26) evaluable questionnaires | Measurement tools: Questionnaire developed by research team and STAI-S for psychological morbidity; Patient satisfaction was greater with video call due to confidence in their physician ($p = 0.006$), efficiency ($p = 0.003$), and punctuality ($p = 0.003$), saving time ($p < 0.0001$) and cost ($p < 0.0001$); Physical exam in 2 vs. 8 visits (9% vs. 31%), prescriptions in 9% vs. 50% of patients, referrals to other professionals in 5% vs. 12% | Very low |

**Table 2.** *Cont.*

| Author and Country | Patient Characteristics | Trial Type and Intervention | Sample Size (Patients per Group) | Outcome | Evidence Certainty |
|---|---|---|---|---|---|
| Beaver et al., 2017 [25]; Dixon et al., 2018 [26]; Beaver et al., 2020 [27] England | Patients with hysterectomy for stage I endometrial cancer; Only 4% had RT | Non-inferiority RCT; Gynecology oncology nurse-led specialist TFU vs. traditional HFU, appointments every 3 or 4 months for 2 years post-treatment followed 6–monthly and annually up to 3–5 years; TFU was on same schedule as HFU; Questionnaires at baseline and immediately after appointments by mail | 259 (129 vs. 130) randomized; 217 (111 vs. 106) analyzed; 211 (105 vs. 106) responded | Psychological morbidity was non-inferior ($33.0 \pm 11.0$ vs. $35.5 \pm 13.0$); Patient satisfaction with information: OR = 0.9, 95% CI 0.4 to 2.1, $p = 0.83$; Patient satisfaction with service: $9.2 \pm 1.5$ vs. $8.9 \pm 1.7$, 95% CI $-0.5$ to 0.3, $p = 0.58$; Recurrence rate of 4%, 5 from each group, all symptomatic and presented as interval events reported to general practitioner or nurse specialist between scheduled appointments; Time from randomisation to diagnosis of recurrence: TFU median 307 days, range 48–662 days; HFU 172 days, range 99–436 days; Cost analysis: no difference at 6 or 12 months; TFU more likely to have appointments on time ($p < 0.001$) and thorough ($p = 0.011$) No statistically significant differences for being able to ask questions, having questions answered, feeling anxious prior to appointments, or feeling reassured; HFU more likely to be kept waiting ($p = 0.001$), and indicated nurse was less likely to be familiar with their case ($p = 0.005$); No significant difference in QoL. | Moderate |
| Viers et al., 2015 [28] USA | Patients with radical prostatectomy $\geq 90$ days and undergoing surveillance, no active urologic concerns requiring physical examination as determined by pre-visit phone call | Equivalence RCT; Video visit (at home or work) vs. office visits by urologist for one visit; Urologists completed a 12-point questionnaire at the conclusion of each visit | 70 (34 vs. 36) randomized; 55 (28 vs. 27) completed the study | Measurement tools: Questionnaire with 7-point Likert scale (1 = strongly agree, 7 = strongly disagree); 100% of video patients and 96% of office visit patients agreed they would meet with their provider in the same setting again; When considering cost, 83% and 59% would choose remote encounter for subsequent visit; No difference in patient trust of the provider (1.0 vs. 1.0), perception of visit confidentiality (1.1 vs. 1.0), or ability to share sensitive/personal information (1.3 vs. 1.0); Similar perceived efficiency (2.1 vs. 1.4), quality of education provided (1.3 vs. 1.4), and overall satisfaction with the encounter (1.2 vs. 1.1); High level of urologist satisfaction in both groups, no difference in quality of medical history, therapeutic management, or perceived patient satisfaction; Different distribution of missed visits: video had 3 technical, 1 canceled, 2 medical, and office visits had 9 late or no show. | Very low |

**Table 2.** *Cont.*

| Author and Country | Patient Characteristics | Trial Type and Intervention | Sample Size (Patients per Group) | Outcome | Evidence Certainty |
|---|---|---|---|---|---|
| Chan et al., 2015 [29] Australia | Chemotherapy patients excluding those with RT or on clinical trials | Non-randomized study; Teleoncology (by general physicians, chemotherapy-proficient nurses, allied health professionals, pharmacist) vs. in-person; Both groups supervised by same medical oncologists | 89 vs. 117 | Serious adverse effects: palliative 5.4% vs. 15%, curative /adjuvant 2.9% vs. 3.6%; Grade 3/4 toxicity in palliative patients: neutropenia 21% vs. 23%, diarrhea 0% vs. 12%, neuropathy 8.8% vs. 0%, fatigue 0% vs. 1.8%, other 8.8% vs. 21%; Grade 3/4 toxicity in curative patients: neutropenia 34% vs. 13%, nausea and vomiting 0% vs. 3.3%, diarrhea 1.8% vs. 1.7%, neuropathy 0% vs. 3.3%, fatigue 0% vs. 6.7%, other 16% vs. 30%; Hospital admissions: palliative 36% vs. 43%, curative 15% vs. 27%; Dose intensity: palliative 97.4% vs. 98.2%, curative 84.4% vs. 88.1%. | Very low |
| Hamilton et al., 2019 [30] Australia | Patients deemed suitable by radiation oncologist or referring specialist; Both new and follow-up appointments eligible | Single-arm study; Tele-radiation oncology program | 311 charts were audited; Survey sent to subset of 231 patients; 106 responses | Survey response rate of 106/231 (46%); 55% preferred telehealth for future appointments, 1% face-to-face, 35% mixed (telehealth and face-to-face), 9% unknown; 80% or more strongly agreed ($\geq$90% agreed or strongly agreed) they could hear the doctor clearly, felt privacy and confidentiality were respected, could ask questions easily, felt it was easy to establish rapport, and thought diagnosis and treatment options were adequately explained; 68% strongly agreed and 15% agreed that they felt reassured when there was a nurse or local doctor present. | Very low |
| Jue et al., 2017 [31] USA | Patients referred to surgical oncologist | Single-arm study; Visit via video from local centre with centralized surgical oncologist who directed oncology treatment, physical exam by nurse practitioner under video supervision of surgical oncologist; Only surgery itself (if needed) was done in-person by surgical oncologist; Single surgeon had medical oncology background | 296 (755 visits) | Reduction in patient travel distance by 80.7% (213,008 miles), saved system $155,627 as patients are normally reimbursed for travel expenses; 86% of patients believed care was more accessible; Average satisfaction scores 4.4/5 to 4.7/5 for most categories; 4.2/5 average for video being more cost and time efficient; 3.8/5 average preference for next visit to be in-person; 3.6/5 average score for believing they would have received better in-person. | Very low |

**Table 2.** *Cont.*

| Author and Country | Patient Characteristics | Trial Type and Intervention | Sample Size (Patients per Group) | Outcome | Evidence Certainty |
|---|---|---|---|---|---|
| Li et al., 2016 [32] China | Lung cancer patients with chronic post-surgical pain after surgery without postoperative complications | Non-randomized comparative study; Remote pain intervention (smartphone or internet) vs. conventional care (weekly in outpatient clinic) | 81 (41 vs. 40) | Measurement tools: SF-36 for QoL at 1 and 3 months after therapy; Similar QoL between groups, $p > 0.05$; Remote group had higher satisfaction (90.2% vs. 72.5%), $p < 0.05$. | Very low |
| >Verma et al., 2015 [33] UK | Men who had RT for localized low to medium risk prostate cancer | Single-arm study; Remote telephone follow-up by radiographer | 134 had first review at 6 months; 69 at 12 months; 9 at 24 months | 88/134 patients (66%) returned questionnaires at 6 months; 77% reported telephone follow-up as more convenient, 3% as not more convenient, 8% no preference; 9% would have preferred an in-person visit with a radiologist and 15% would have preferred an in-person visit with a doctor; For future follow-up, 76% preferred phone, 6% preferred in-person, 7% no preference. | Very low |
| Patil et al., 2018 [34] India | Adult patients with intermediate- to high-grade (grade II-IV) glioma on adjuvant TMZ for 2+ cycles | Single-arm study; Video (VF) vs. clinic follow-up (CF) in same patients; VF on day 24 from previous CF to determine if patients needed continuation of TMZ, supportive medications, imaging, molecular testing, and rehabilitation; In-person exam about 4 days later to determine agreement of decision; Two groups of 3 clinicians did assessment; Patients randomized as to which group did assessment | 65 | Concurrence in decision to administer TMZ was 100%; Patient satisfaction rate (somewhat or extremely) was 100% post-VF and 98.5% post-CF; Median cost $58.15 US vs. $131.23 US. | Very low |

**Table 2.** *Cont.*

| Author and Country | Patient Characteristics | Trial Type and Intervention | Sample Size (Patients per Group) | Outcome | Evidence Certainty |
|---|---|---|---|---|---|
| Smrke et al., 2020 [35] | Patients with sarcoma | Non-randomized comparative study; Telemedicine vs. face-to-face; Anonymous patient experience survey and clinician survey | 108 patient surveys returned (70 vs. 34); 18 clinician surveys | Patient satisfaction: 8.99/10 vs. 8.35/10; 80% indicated they would like at least some future appointments to be by telemedicine for reasons of travel time (42%), travel expense (20%), and convenience (30%); 48% would not want bad news over the phone; Of those who preferred face-to-face, 42% felt it was more reassuring and 20% felt the treatment plan was clearer; 78% clinicians found appointments shorter by phone; 89% felt it did not increase workload, but affected ability to perform exams sometimes (44%) or often (17%); Most clinicians favoured telemedicine for active surveillance or stable-dose oral medication. | Very low |
| Rodler et al., 2020 [36] Germany | Patients with advanced genitourinary cancer and systemic therapy switched to virtual treatment (phone or video) where possible due to COVID-19 pandemic; Visits limited to therapy application | Single-arm study; Survey of all patients during one week by email, phone, or in-person | 101 patients; 92 responded | Majority of patients valued continuation of therapy higher than COVID-19 prevention measures; 77.2% were unwilling to postpone a staging exam; 44.6% were afraid of progression (61.8% chemotherapy; 33.3% immunotherapy) and did not want to delay or interrupt treatment; Acceptance of virtual discussion of staging results and therapy decisions: median 8/10 (IQR 5–9); Referral to secondary care oncologists for therapy: median 2.5/10 (IQR 0–6.75); Preference for telehealth beyond the pandemic: median 4/10 overall (IQR 2–7; 3/10 for immunotherapy and 5/10 for chemotherapy); 62.6% preferred in-person care after the pandemic; High acceptance of external laboratory controls (60.9%) but lower acceptance for online visit management (48.9%), remote treatment planning (44.6%), and referral to secondary care oncologists (17.4%). | Very low |

**Table 2.** *Cont.*

| Author and Country | Patient Characteristics | Trial Type and Intervention | Sample Size (Patients per Group) | Outcome | Evidence Certainty |
|---|---|---|---|---|---|
| Layfield et al., 2020 [37]; Triantafillou et al., 2020 [38] USA | New (6%) or returning (94%) otolaryngology patients who previously had telemedicine visits; 73% malignant, 9% premalignant | Single-arm study; Video-based telemedicine visits | 100 qualitative comments from subset of 56 patients | Mean scores ± SD: Usefulness 6.10 ± 0.50; Ease of use 6.21 ± 0.13; Effectiveness 6.20 ± 0.60; Reliability 4.86 ± 0.84; Satisfaction 6.29 ± 0.32; Concerns were related to limitation on physical exams and lack of touch; 29% required technical assistance from family or caregiver; 32% expressed relief they could receive care during the pandemic; 25% indicated telemedicine was more convenient. | Very low |

Abbreviations: CI = confidence interval, COVID-19 = Coronavirus disease of 2019, CSQ = Client Satisfaction Questionnaire, HFU = hospital follow-up, IQR = interquartile range, OR = odds ratio, QoL = quality of life, RT = radiotherapy, SD = standard deviation, SF-36 = 36-item Short Form Health Survey, STAI-S = State Trait Anxiety Inventory, TFU = telephone follow-up, TMZ = temozolomide, vs = versus.

An RCT found that videoconference delivery of psychosocial pain management resulted in higher rates of session completion, was noninferior for pain severity and pain interference, and was more feasible than in-person management [21,22]. Both groups had similar patient burden and engagement, and degree of acceptability. A strength of this study was that the videoconference group was given a tablet (iPad) with a data plan; these were also given to those in the in-person group if needed to access the study website for self-assessment and to enter content preferences.

In the NCT-MOBILE RCT, those having video follow-up after systemic therapy for solid tumours had significantly greater satisfaction with the interaction and more confidence in the physician, and found it had greater efficiency, punctuality, time saving, and lower cost, compared to those having standard in-person visits [23,24]. Difficulties in the first appointment were mostly due to software incompatibility and internet connections that were resolved for the second appointment.

The ENDCAT trial randomized patients to either telephone or traditional hospital follow-up after hysterectomy for endometrial cancer [25–27]. Differences in patient satisfaction, quality of life, being able to ask questions, having questions answered, feeling anxious prior to appointments, feeling reassured, and cost were not significant. Telephone appointments were more likely to be on time and thorough. Recurrence was the same in both groups (4%) and all were symptomatic; patients with recurrence were excluded from further follow-up.

The Mayo Clinic trial randomized 70 men after radical prostatectomy with no urologic concerns to either video or in-office follow-up for one visit [28]. Patients reported no difference or similar trust in the provider, education provided, satisfaction, visit confidentiality, and ability to share personal information. There were significantly lower costs to the patients for video visits.

Nine non-randomized studies were small (65–296 patients) and in most cases the comparative group, if any, was not equivalent [29–38]. Three of the studies had patients attend a local clinic with nurses or non-specialist physicians and remote contact to specialists [29–31]. Patients were generally satisfied, although some preferred a mixture of traditional and remote care; transportation costs were reduced and care for some was more accessible. The narrow disease-specific stage-specific studies suggest virtual care may be suitable in well-defined specific situations [32–34]. As is generally the case for non-randomized studies, the risk of bias is high and quality of evidence from these trials is considered low to very low. They do suggest, however, that most patients were satisfied with virtual care.

Three recent studies of virtual care implemented due to the COVID-19 pandemic were included. The Smrke et al. study reported higher patient satisfaction with telephone consultations than in-person visits [35]. Most patients (80%) preferred at least some future appointments to be by telemedicine for reasons of travel time (42%), travel expense (20%), or convenience (30%). Clinicians reported phone consultation sometimes (44%) or often (17%) affected the ability to perform examinations, but most favoured this for active surveillance or stable-dose oral medication. The comparison in-person group was not considered equivalent. Two studies involved surveys or questionnaires of patients who received virtual care (no comparison groups). One survey indicated that most patients with advanced genitourinary cancer wanted therapy continued despite COVID-19 risks [36]. They accepted virtual discussion of staging results and therapy decisions, but had lower acceptance of referral to secondary care oncologists for therapy. Another survey showed that video-based telemedicine was rated high for usefulness, effectiveness, and satisfaction, but lower on reliability due to limitations on physical examinations in otolaryngology patients [37,38].

### 3.3. Ongoing, Unpublished, or Incomplete Studies

Five ongoing studies were found in the literature search and details are summarized in the Supplementary Material (Table S2) [39–43].

## 4. Discussion

This systematic review determined that oncology studies with direct comparison between virtual and in-person care are limited and generally provide low to very-low quality evidence, with the exception of RCTs that studied very specific situations: genetic counselling and endometrial cancer follow-up. While there is intense interest in understanding where virtual care may be at least equivalent to in-person care during active cancer management (post surgical care, chemotherapy radiation therapy), this review found little published evidence that directly addressed this aspect of cancer care.

In the general field of psychological counselling, virtual or remote counselling has been reported in several reviews to be equivalent to in-person counselling [44–49]. While evidence with cancer patients in the current review is limited, studies suggest virtual counselling and in-person counselling may have similar effectiveness in treating anxiety, stress, depression, and adjustment issues [5,6]. A survey of patients and therapists involved in psycho-oncology counselling video-consults found that approximately one-half of the patients expressed preference for video-consults in the future for a portion of sessions [8]. While therapists were willing to provide video counselling if requested, they preferred in-person sessions, especially for more complex issues. These studies indicate that individual situations and patient preferences need to be considered.

In the area of genetic counselling, it was reported that telephone counselling was noninferior to usual care for all outcomes (knowledge, decision conflict, cancer distress, perceived stress, genetic counseling satisfaction) and was more convenient [14–17]. On the negative side, there was lower perceived support and emotional recognition. Some studies also found lower costs to the patient or system for virtual counselling. It should be noted that genetic counselling is usually limited to a small number of appointments, usually a pre-test appointment and sometimes a follow-up appointment for patients choosing to undergo genetic testing. This model typically involves few interactions between patients and providers with little long-term follow-up, in contrast to many other areas of cancer care. As such, results from this study may not be transferable to other types of clinician-patient interaction. The combined evidence from all the counselling studies suggests that visual cues (body language, expression) available with in-person or video sessions may be more important for in-depth counselling, and therefore, video methods have advantages over telephone use.

Another area where virtual care appears to have extensive use is in long-term follow-up for asymptomatic patients with endometrial cancer. The ENDCAT trial, evaluated as high quality, was conducted in patients with stage I endometrial cancer [25–27]. It found high patient satisfaction and noninferior psychological morbidity for telephone versus in-person follow-up. All recurrences were symptomatic and detected between scheduled virtual or in-person visits, suggesting that virtual follow-up in early-stage endometrial cancer does not place patients at increased risk. Most other studies measured outcomes of patient satisfaction, feasibility, and cost, and one studied pain management, but they did not include long-term outcomes. Overall, the results were consistent in showing feasibility and patient acceptance of virtual care. It remains unknown if these findings in the area of endometrial cancer can be generalized to follow up of asymptomatic patients in other cancer disease sites.

*Limitations*

First, since our literature search period is from 2015, some relevant earlier publications may have been missed. However, virtual care technology has developed rapidly in recent years, and the pace of change has accelerated with the onset of the global pandemic in 2020. Older studies by phone have been supplanted by videoconferencing and new delivery platforms. This changing environment makes a traditional model of data collection and evidence generation very challenging. One option is to consider regional demonstration projects with common data collection elements that map to traditional quality frameworks. Second, although ten RCTs met our pre-planned study selection criteria, eight of them

were of low to very low quality. It is realized that the conduct of RCTs testing virtual care strategies is susceptible to high risks for bias for the measurement of patients reported outcomes. Third, only one eligible study reported recurrence rate, which we consider to be a critical outcome. Even in ongoing trials from the clinicaltrial.gov search (Table S2), no trials will report recurrence, survival, or other long-term outcomes.

## 5. Conclusions

According to current but limited evidence, virtual care does not appear to be inferior to in-person care when considering patient satisfaction. The published literature does support the use of virtual platforms for counselling interventions that include genetics and psychosocial care. It is clear that more work is required to understand the disease, patient, and system factors that a virtual care approach can reliably support.

The COVID pandemic has necessitated rapid adoption of virtual cancer care without a strong evidence base or systematic stakeholder engagement around the perceived benefits and limitations of this approach, especially during active cancer treatment. The current environment offers a unique opportunity for gathering data and evidence at both the patient and cancer system levels. RCTs examining evidence-based evaluation of virtual care are lacking, although the post pandemic phase may provide some opportunity as the pendulum of balance between in-person and virtual care shifts. Given the investment in infrastructure and rapid implementation of virtual care, there is an important opportunity to address some of the critical issues that have not been comprehensively studied to date that include treatment outcomes and patient safety.

**Supplementary Materials:** The following are available online at https://www.mdpi.com/article/10.3390/curroncol28050301/s1, Table S1: Literature Search Strategy (Databases); Table S2: Ongoing trials; Table S3: Summary Table for Risk of Bias Assessment for Randomized Controlled Trials.

**Author Contributions:** Conceptualization, S.S. and J.S.; methodology, S.S., G.G.F., X.Y. and J.S.; investigation, X.Y. and G.G.F.; writing—original draft preparation, all authors; writing—review and editing, all authors. All authors have read and agreed to the published version of the manuscript.

**Funding:** This research was conducted by the Program in Evidence-Based Care (PEBC) at McMaster University, a provincial initiative of Ontario Health (Cancer Care Ontario) supported by the Ontario Ministry of Health (OMH). All work produced by the PEBC is editorially independent from the OMH.

**Institutional Review Board Statement:** Not applicable.

**Informed Consent Statement:** Not applicable.

**Data Availability Statement:** Data used in this study can be found in the included original studies.

**Acknowledgments:** The authors would like to thank Faith Maelzer for searching the clinicaltrials.gov database of ongoing trials and conducting a data audit and Megan Smyth for assisting in assessing the risk of bias for RCTs.

**Conflicts of Interest:** The authors declare no conflict of interest.

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
