# Peer review of "Virtual Care in Patients with Cancer: A Systematic Review"

_curroncol, doi:10.3390/curroncol28050301_

Round 1

Reviewer 1 Report

This timely review summarises the evidence for video or phone consultation with cancer patients across different treatment types. Overall the paper is well conceived and written. I have only a few major suggestions and some minor corrections.

  1. It is not clear if risk of bias was double coded? Can the authors clarify.
  2. In Figure 1, 10,399 papers were screened and 10,108 were excluded, but reasons for exclusion were only provided for 75 papers? What about the others? Also in the last box in Figure 1, the word "competed" should be replaced with "completed".
  3. I understand the author's decision to go for a narrative synthesis of the literature, but I do think a Table listing the outcomes investigated and the number of papers that provide a positive, negative or neutral benefit for telehealth would be useful summary for the reader. 
  4. p.5, line 126, ...due to lack of non-verbal communication and (add non-recognition of) signs of distress.
  5. Discussion, p.12, second line   ....limited and (add space) generally...
  6. p.13, last two paragraphs are identical - one needs to be removed!
  7. p.13 line 310, ...finding(add s) in the area...
  8. p.14, line 240, ...most favoured this for actively surveillance (replace or with of) stable-dose oral medications. 
  9. Conclusions, line 331  ...when considering  (reduce space) patient satisfaction...

Author Response

Dear Reviewer 1, Thank you very much for your careful review and very helpful comments. Please see our detailed responses on each comment below in italic blue words.

  1. It is not clear if risk of bias was double coded? Can the authors clarify.

---We have added more details on lines 85-87 and lines 113-114.

  1. In Figure 1, 10,399 papers were screened and 10,108 were excluded, but reasons for exclusion were only provided for 75 papers? What about the others? Also in the last box in Figure 1, the word "competed" should be replaced with "completed".

---It appears the figure was misinterpreted. It has been revised to be clearer. 10108 papers were determined to be off topic based on screening titles and/or abstracts only, and 75 additional publications were guidelines, reviews, technical documents addressing the topic but not clinical studies for inclusion in the review itself. After titles and abstracts screening, there were 216 papers for which the full text needed to be reviewed to determine inclusion or exclusion. We provided the excluded reasons for 177 papers after the full-text review.

  1. I understand the author's decision to go for a narrative synthesis of the literature, but I do think a Table listing the outcomes investigated and the number of papers that provide a positive, negative or neutral benefit for telehealth would be useful summary for the reader. 

---Studies meeting the inclusion criteria for the systematic review have been summarized in Tables 1 and 2, and readers are referred to these at the start of each subsection of the Results. We do not believe it’s possible to group studies as positive, negative and neutral because this type of evaluation depends on which outcome is looked at, and even for a specific outcome it varies according to patient characteristics within the study.

  1. p.5, line 126, ...due to lack of non-verbal communication and (add non-recognition of) signs of distress.

---We have added it on line 131.

  1. Discussion, p.12, second line   ....limited and (add space) generally...

---Many thanks for catching it. We have revised it.

  1. p.13, last two paragraphs are identical - one needs to be removed!

---This has been addressed.

  1. p.13 line 310, ...finding(add s) in the area...

---This has been corrected.

  1. p.14, line 240, ...most favoured this for actively surveillance (replace or with of) stable-dose oral medications. 

---This is correct as written.

  1. Conclusions, line 331  ...when considering  (reduce space) patient satisfaction...

---This sentence has been revised.

Thank you very much again for your reconsideration and time.

We are looking forward to hearing from you soon.

Kind regards,

All the co-authors

Reviewer 2 Report

The present study intends to review the share of virtual care in oncology due to the ongoing covid-19 pandemic.

The study is well designed.

My big concern is that the authors did not show evidence of the novelty of the study. They need to search for similar studies and thus show what this study adds new to the evidence. The authors are best advised to cover this to add more value to the paper.

Author Response

Dear Reviewer 2,

Thank you very much for your careful review and the comment. As indicated in the introduction and inferred in Figure 1 (box indicating 75 guidelines, reviews and technical documents were identified for other parts of the project), this systematic review is based on a larger report. Due to space limitations, we limited the current publications to the clinical studies. Eight systematic reviews of partial relevance are reported in the full document on the Ontario Health website, which covered narrower topics, did not focus on cancer, missed several of the studies that we found, or were based on non-comparative studies. Therefore, we determined that our own review should proceed. In response to your comment, additional information has been added to sections 2.1 and 2.2 of the Materials and Methods sections.

Thank you very much again for your reconsideration and time.

We are looking forward to hearing from you soon.

Kind regards,

All the co-authors